# A Review of Contemporary Guidelines and Evidence for Wide Local Excision in Primary Cutaneous Melanoma Management

**DOI:** 10.3390/cancers16050895

**Published:** 2024-02-23

**Authors:** Sophie E. Orme, Marc D. Moncrieff

**Affiliations:** 1Norfolk & Norwich University Hospitals NHS Foundation Trust, Norwich NR4 7UY, UK; sophie.orme@nnuh.nhs.uk; 2Norwich Medical School, University of East Anglia, Norwich NR4 7TJ, UK

**Keywords:** melanoma, margin, excision, recurrence, survival

## Abstract

**Simple Summary:**

The vast majority of patients who present with a primary cutaneous melanoma can be cured by surgery alone. Although a wider local excision margin around the primary tumour may in theory maximize the chance of cure, the result is a larger wound that often requires a more complex operation to close, as well as greater risk of surgical complications, morbidity, and higher associated healthcare costs. Despite several previous studies, we have yet to reach agreement internationally over what excision margin is optimal. This paper reviews the evidence for current guidelines for wide local excision margins; explores the challenges of extrapolating the findings of previous randomised trials into clinical practice within the rapidly evolving landscape of modern melanoma management; and finally discusses the potential of the actively enrolling MelMarT-II trial to provide a definitive answer to the question: how wide is wide enough?

**Abstract:**

Surgical wide local excision (WLE) remains the current standard of care for primary cutaneous melanoma. WLE is an elective procedure that aims to achieve locoregional disease control with minimal functional and cosmetic impairment. Despite several prospective randomised trials, the optimal extent of excision margin remains controversial, and this is reflected in the persistent lack of consensus in guidelines globally. Furthermore, there is now the added difficulty of interpreting existing trial data in the context of the evolving role of surgery in the management of melanoma, with our increased understanding of clinicopathologic and genomic prognostic markers leading to the often routine use of sentinel node biopsy (SNB) as a staging procedure, in addition to the development of adjuvant systemic therapies for high-risk disease. An ongoing trial, MelMarT-II, has been designed with the aim of achieving a definitive answer to guide this fundamental surgical decision.

## 1. Introduction

The rising incidence of melanoma internationally has been described by some as an “epidemic”, with the global burden of cases estimated to increase by a further 50% by 2040 [1]. Melanoma is one of the most common cancers in young adults, especially young women and, although the greatest incidence is seen in those over 75 years of age, approximately 45% of patients are diagnosed before the age of 65 years old [1,2]. In parallel, melanoma accounts for over 80% of skin cancer deaths, with one of the highest mortality rates in young and middle-aged adults for all adult-onset malignancies, and as such is a significant, growing socio-economic challenge for many countries [1,3].

With recent advancements including the advent of adjuvant systemic therapies, we have entered a new era in the management of melanoma. In the midst of these promising developments, it may easily be overlooked that, for the majority of patients who present with a primary cutaneous melanoma, the definitive treatment is surgery alone. Surgery was the only treatment modality required for 90% of patients diagnosed with Stage II primary cutaneous melanoma in England between 2013 and 2020 [4]. Indeed, 75–88% of these patients will survive for 10 years or longer [5]. Therefore, the optimisation of surgical decision making is crucial not only for prognosis, but increasingly for survivorship issues such as improving quality of life outcomes.

The foundation of the surgical management for melanoma is wide local excision (WLE). WLE was first proposed by the physician William Handley, writing on his observations of a single case of metastatic melanoma in the *Lancet* in 1907. Since then, WLE margins of 5 cm of subcutaneous tissue down to the level of muscle fascia, along with the radical removal of lymph nodes, became the universally accepted treatment for over 50 years [6,7]. The purpose of WLE is to achieve locoregional disease control by removing any local micrometastases as well as any genotypically unstable cells harboured within otherwise phenotypically normal surrounding tissue of the dermis and superficial lymphatics.

Locoregional relapse clinically manifests as in-scar recurrence or in-transit metastases (ITMs—also known as satellite metastases), often in combination with regional lymphadenopathy. Clinically, the patient presents with dermal intralymphatic deposits of tumour of varying size and quantity, interposed between the primary tumour site and the draining lymphatic basin (Figure 1). The development of ITMs heralds a poor prognosis for the patients, with 50% of patients dying of their melanoma within three years of confirmation of the diagnosis [8]. Furthermore, the poor prognosis associated with the finding of microsatellites, which are microscopic deposits of melanoma seeded into the dermis by intralymphatic spread, on histological examination after WLE, supports the rationale that wider excision margins to capture these cells might reduce the likelihood of local recurrence (LR). The incidence of microsatellites in primary cutaneous melanomas ranges between 3 and 19% and their adverse impact on both recurrence-free and disease-specific survival is significant enough that their presence upstages the patient to Stage III in the AJCC 8th classification system [5,9,10,11].

Excision of the primary with a WLE margin that is inadequate to capture surrounding genotypically abnormal cells (also termed field cells) may lead to local persistence of disease, as they then may act as precursor tumour cells evolving further genetic aberrations which confer invasive and metastatic potential over time, eventually resulting in locoregional relapse despite a clear histological margin at the time of WLE. In an analysis of 19 acral melanoma specimens, North et al. found that field cells were detectable using fluorescent in situ hybridisation (FISH) in 84% (16 out of 19) of cases, typically found in an asymmetrical distribution extending from the histological intraepidermal margin of the melanoma by a mean distance of 4.5 mm (range 2.0–9.0 mm) in invasive melanomas and 6.1 mm (range 1.5–12.5 mm) for in situ melanomas [12]. The presence of field cells may represent a plausible mechanism for local persistence and eventually locoregional relapse by contiguous spread in other subtypes of melanoma, but has yet to be established as their genomes typically contain a lower number of DNA amplifications than the characteristically high level seen in invasive acral melanomas [13]. This is further supported by work undertaken in the Melanoma Institute of Australia, which found that for patients with melanomas ≤ 2 mm thick, histologic margins < 8 mm were associated with a higher rate of local recurrence, whereas histologic margins < 16 mm in those with tumours > 4 mm thick had worse local disease-free survival (HR, 2.41; *p* = 0.01) [14,15]. These histologic margins correspond to measured excision margins of 1 and 2 cm, respectively, assuming a 20% shrinkage rate of formalin-fixed specimens; however, the correlation between clinical and histologic margins is highly variable [16].

It is interesting that North et al. found no correlation between the extent of field cells and the tumour thickness (R^2^ = 0.001), as the risk of LR is directly proportional to the T-stage of the primary [17]. This is reflected in the current international guidelines where the width of recommended margins is proportionate to the invasive depth (Breslow thickness, BT) of the primary melanoma, i.e., smaller for early invasive melanoma and greater for thick primaries, as illustrated in Table 1.

However, several prospective randomised controlled trials (RCTs) and subsequent meta-analyses have yet to provide definitive evidence that narrower margins are associated with poorer survival outcomes, and consequently, the recommended WLE margins have been reduced considerably since Handley’s initial report. Furthermore, long-term follow-up data have shown that wider margins are significantly associated with poorer quality of life outcomes, including increased risk for reconstructive surgery; adverse surgical events; risk of chronic pain; prolonged hospital stay and rehabilitation as well as increased associated healthcare costs [25,26,27]. There remains growing concern worldwide among many surgical oncologists that the width of recommended excision margins continues to result in an unjustifiable degree of morbidity excessive for the majority of patients presenting in the absence of indicators of more biologically aggressive disease.

Despite over a century of debate, there is persistent ambiguity reflected in the current national guidelines for excision margins for primary cutaneous melanoma. A definitive answer is needed. In this paper, we summarise and review the evidence from RCTs completed to date, evaluating a range of excision margins. We also discuss how the rapid developments in our understanding and management of this disease have further challenged the integration of these findings into contemporary practice and highlight why new evidence established within this landscape is needed. Finally, we will focus on the potential contribution of the MelMarT-II trial to future practice and areas where further research may still be required.

## 2. Literature Search and Selection

A literature search was performed on Pubmed, Embase, and Cochrane Central Register of Controlled Trials (CENTRAL) using the search strategy: (melanoma) AND (wide local excision OR margin). For PubMed and Embase filters for clinical trials, randomised controlled trials and systematic reviews were used. There were no restrictions based on language, publication status, or year. Studies were excluded if they were repeat reports of a previously published trial or if they were concerned only with melanoma in situ. Articles were screened for relevance of their title and abstract. The reference lists of relevant articles and systematic reviews were then hand-searched.

## 3. Is There Prognostic Benefit of Wide Margins Compared to Narrow Margins?

To date, there has been a total of seven completed RCTs (shown in Table 2), which includes a recently published feasibility report for the actively enrolling MelMarT-II trial [28,29,30,31,32,33,34]. A further six meta-analyses, including a Cochrane review, have concluded that the current evidence is insufficient to identify the optimal excision margins [7,35,36,37,38].

### 3.1. Melanomas < 2 mm Thick

For decades following Handley’s initial report, WLE with margins of 3 to 5 cm was recommended and performed routinely until as recently as the 1970s. However, following Breslow’s seminal work elucidating tumour thickness as a prognostic marker for local recurrence, it was postulated that locoregional control of thin lesions (pT1-pT2) might be sufficiently achieved through more conservative margins [44]. Evidence supporting this approach was provided from a retrospective analysis, and the first RCT, comparing 1 cm and 3 cm excision margins, was undertaken by the World Health Organization (WHO) Melanoma Group and published in 1988 [28,45]. After extended follow-up, they found that there was no statistically significant difference in either the disease-free survival (DFS) or overall survival (OS) (85.2% vs. 87.3%) for participants that were treated with a 1 cm compared to a 3 cm WLE margin at 7.5 years [39,40]. Four patients, clustered within the narrow 1 cm excision arm, had an LR as a first relapse. There were a number of limitations, not least that the protocol called for excisions to be undercut by an additional 1–2 cm within the subcutaneous fat, resulting in ambiguity in the true width of the margins. Furthermore, the number and characteristics of those lost to follow-up is unclear. Nevertheless, given the observation that all four events of LR occurred in those with melanomas thicker than 1 mm, they concluded that excision with narrower margins is a safe and effective procedure for those with primary melanomas thinner than 1 mm [39,40].

A further two randomised trials comparing 2 cm to 5 cm excision margins followed. The European Trial undertook the comparison for 326 patients with primary tumours less than 2.1 mm thick [30]. There were almost twice as many female patients (*n* = 204) than male (*n* = 122) but the two groups were comparable for prognostic characteristics including sex. After a median follow-up time of 16 years, 40 patients (12%) were lost to follow-up and a further 36 were not evaluable for DFS due to missing data. They found no significant difference in 10-year OS (87% for 2 cm vs. 86% for 5 cm, *p* = 0.56) or 10-year DFS (85% for 2 cm vs. 83% for 5 cm, *p* = 0.83). LRs occurred in one patient in the 2 cm arm compared to four patients in the 5 cm arm (overall LR rate = 1.5%) with no significant difference observed between groups [30]. The Swedish I trial enrolled 989 patients from six national regions with melanomas between 0.8 and 2.0 mm in thickness. Once again, groups were comparable for gender, age, tumour site, subtype, and thickness [29]. They found no statistically significant difference in either OS (HR 0.96; 95% CI 0.75–1.24; *p* = 0.77; median follow-up 11 years) or recurrence-free survival (RFS) (HR 1.02; CI 0.80–1.30; *p* = 0.88; median follow-up eight years). LR as a first relapse was rare, with five events in total observed (overall rate = 0.5%, one event in the 2 cm arm and four events in the 5 cm arm). There was no significant difference in LR events between the 2 cm and 5 cm arms (0.2% vs. 1.0%, respectively) or locoregional recurrence (15% vs. 12%, respectively, *p* = 0.22) [41]. Both trials, therefore, concurred that patients with melanomas between 1 and 2 mm thickness could be safely be treated with a WLE margin of 2 cm, and wider margins were unnecessary [29,30,41].

The Intergroup trial enrolled a further 486 patients with intermediate thickness (1–4 mm) melanomas of the trunk or proximal extremity, from multiple institutions in four countries (United States, Canada, Denmark, and South Africa) and randomised them to either 2 cm or 4 cm margins. After a median follow-up of six years, the authors found no significant difference in either OS (79.5% vs. 83.7%) or LR rate (0.8% vs. 1.7%) for the 2 cm margin patients vs. the 4 cm margin patients, respectively [31]. In long-term analysis, 468 patients (6/244 lost from the 2 cm arm, 12/242 lost from the 4 cm arm) were followed up to a median of 10 years [26]. The rate of LR as a first relapse was similar in both groups (0.4% and 0.9% for 2 and 4 cm margins, respectively) and the trial found no significant differences in the 10-year disease-specific survival (DSS) between those who underwent a 2 cm compared to a 4 cm excision margin (DSS 70% vs. 77%, *p* = 0.074). The authors concluded that the 2 cm margin is safe for those with melanomas < 4 mm in thickness; however, in subsequent analyses, some have interpreted this nearly significant finding as an indication that there may still be a potential adverse effect of margins ≤ 2 cm [26,37].

Furthermore, the group analysed the patterns of metastases in those who developed a local recurrence at any time (*n* = 28) out of the entire study cohort (*n* = 740). They found that in the majority of cases (62%), the next site of relapse occurred at distant rather than regional sites, lending weight to the theory that local recurrences were more likely a manifestation of distant metastasis within the skin or subcutaneous tissue, arising from circulating metastatic melanoma cells, rather than arising from retained primary tumour cells or genetically unstable cells left behind after an inadequate incision. These analyses included those with melanomas of the head and neck and distal extremities who were not randomised and received a 2 cm excision margin, (*n* = 272), and for whom the LR rate was found to be much higher [26]. The Intergroup trial also found that there was significantly greater treatment morbidity and length of hospital stay in the 4 cm margin group. With this in mind, the authors concluded that for patients with intermediate thickness melanomas, adopting a 2 cm WLE margin would reduce the risk of associated treatment morbidity, without any compromise to prognosis and an acceptably low risk of LR (<1%) [26,31,42].

### 3.2. Melanomas > 2 mm Thick

Of the first four trials undertaken, only the Intergroup trial included patients with primaries thicker than 2 mm, and this subgroup was smaller (213/486 patients, 44%) [31]. Furthermore, none of the preceding trials had included patients with melanomas of > 4 mm thickness. Therefore, two further trials were undertaken in patients with melanomas ≥ 2 mm thick, first by the UK Melanoma Study Group (UK MSG) comparing 1 cm with 3 cm margins, and subsequently by the Swedish Melanoma Study Group comparing 2 cm to 4 cm margins. Consistent with preceding RCT evidence, the UK MSG trial found no statistically significant difference in either disease-specific (HR 1.24; 95% CI 0.96 to 1.61; *p* = 0.1) or overall survival (HR 1.07; 95% CI 0.85–1.36; *p* = 0.6) for patients treated with 1 cm and 3 cm excision margins at a median follow-up of five years. However, they did report a statistically significant difference in the locoregional recurrence rate (HR 1.26; 95% CI 1.00–1.59; *p* = 0.05). Although at that time the authors noted a difference in the number of deaths due to melanoma in the 1 cm excision arm (*n* = 128) compared to the 3 cm excision arm (*n* = 105), this was not found to be statistically significant (HR 1.24; 95% CI 0.96 to 1.61; *p* = 0.1) [32]. However, in further analysis of these data, which extended the median follow-up to 8.8 years, Hayes et al. found that the 1 cm margin was associated with a statistically significant reduction in DSS compared to the 3 cm arm (absolute difference 5.95%), leading the authors to conclude that the survival of 1 in 16 patients could potentially be disadvantaged by excision margins < 3 cm at 10 years (HR 1.24 [95% CI 1.01–1.53], *p* = 0.041) [46]. However, this study has significant limitations which bring the authors’ interpretations into serious doubt, most notably that none of the participants were staged using SNB. It is notable that statistical significance for LR was only established when both local and regional nodal recurrences were combined, and it remains possible that the difference in both the locoregional recurrence rate and DSS were instead a consequence of a higher incidence of microscopic locoregional extension in the narrow margin group at the time of diagnosis [47].

In contrast, the Scandinavian trial recruited patients < 75 years of age with melanomas thicker than 2 mm (median BT 3.1 mm), from 53 hospitals in Sweden, Denmark, Estonia, and Norway, and randomised them to either a 2 cm or a 4 cm excision margin. With a cohort of 936 patients, it represents the largest RCT published; however, with changes to clinical practice favouring routine excision with narrow margins towards the end of the enrolment period, the trial was terminated early and the overall accrual did not meet the planned sample size to show equivalency. Clinicopathological features were similar in both groups. At initial follow-up after a median 6.7 years, no significant difference was observed between the two groups for both 5-year OS (65% in both groups, *p* = 0.69) and 5-year RFS (56% in both groups, *p* = 0.82). The number of deaths due to melanoma was also equal (134/465 in the 2 cm arm vs. 138/471 in the 4 cm arm) (HR 0.99; 95% CI 0.78–1.26, *p* = 0.95). Once again, LR as a first event was rare (overall rate 3%), and although there were twice as many occurrences within the 2 cm margin group compared to the 4 cm margin group, this finding did not reach statistical significance (*n* = 20, 4.3% vs. *n* = 9, 1.9%; *p* = 0.06). When nodal metastasis and in-transit metastases were combined into a hybrid endpoint of locoregional recurrence, the outcome was equal (139 vs. 138 events) in the two treatment groups (HR 1.00; 0.79–1.28; *p* = 0.96) [33]. The long-term follow-up was the most complete of all the RCTs, with a median of 19.6 years and <1% loss to follow-up (2/936). Consistent with their initial findings and those of previous RCTs, there was no significant difference observed between the survival curves for either OS (HR 0.98; 95% CI 0.83–1.14; *p* = 0.75) or DSS (HR 0.95; 95% CI 0.78–1.16; *p* = 0.61) [43]. The findings of this study supported the safety and efficacy of the 2 cm excision margin in patients with thick (>2 mm) melanomas, which, at the time of the report, was the current clinical practice in several countries including Australia, the USA, Canada, and the Netherlands. Furthermore, it notably contributed to the revision of the recommended 3 cm margin in patients with a BT of >2 mm to a 2 cm margin by the UK’s National Institute for Health and Care Excellence (NICE) in 2015 [18].

Towards the end of the enrolment period for the Scandinavian trial in 2004, the prognostic significance of sentinel node (SN) positivity had been realised, leading to sentinel node biopsy (SNB) becoming a recommended routine staging procedure for the trial’s patient cohort [48,49]. However, only a small proportion of patients (n = 81 patients, 9%) underwent the procedure, which revealed an SN-positive rate of 44% (23/51 in the 2 cm group and 13/31 in the 4 cm group) [33]. With the significant pace of changes in clinical practice, there emerged a demand for a sufficiently sized RCT with a contemporary design, reflecting the impact of advancements in both the understanding and management of melanoma over the last two decades. A protocol for such a trial, the Melanoma Margins Trial (MelMarT) trial, emerged in 2014.

### 3.3. MelMarT

With the majority of data supporting the hypothesis that primary cutaneous melanomas may be managed with narrow margins with similar safety and efficacy to that of wider margins, yet with potentially substantially less associated morbidity and cost, there remained a lack of any direct comparison between the extent of “narrow” margins previously evaluated, namely 1 cm and 2 cm [7]. The Melanoma Margins Trial (MelMarT) II trial (ClinicalTrials.gov ID: NCT03860883) is an international multicentre, randomised controlled phase III clinical trial, investigating the hypothesis that a 1 cm wide excision margin is non-inferior to a 2 cm wide excision margin for patients with primary, cutaneous, pT2b-4b melanomas [33].

Its design was unique compared to previous RCTs in a number of key criteria reflecting contemporary practice. MelMarT represents the first and only trial sentinel lymph node biopsy as an essential inclusion criterion for pathological staging. Furthermore, it has a pragmatic design, meaning that those patients with positive SNB will be managed according to the treating unit’s local protocol, allowing for the effects of surgical margins to be evaluated within the new, current context of the availability of adjuvant therapies for high-risk disease. In contrast to preceding RCTs, MelMarT was designed as a formal non-inferiority trial; the trial design initially included pT2a melanomas and given the particularly low event rate in this cohort, a sample size of nearly 10,000 patients was required to provide enough statistical power to definitively address the safety and efficacy of 1 cm vs. 2 cm margins. Consequently, it was prudent to conduct a pilot feasibility study to determine if patient recruitment could be achieved at a sufficient rate across multiple centres internationally.

The first phase of the trial, MelMarT-I (ClinicalTrials.gov ID: NCT02385214), recruited 400 patients from 17 centres in five countries between January 2015 and June 2016, and published its feasibility report in 2018. With only one year of follow-up, the data were too immature to report on LR and survival data. Most critically, the results demonstrated the feasibility of the trial to provide a definitive answer to the optimal excision margin for patients with intermediate to high-risk primary cutaneous melanomas, with an encouraging 66% enrolment rate. Furthermore, the report included initial quality of life (QOL) outcomes data measured using the FACT-M questionnaire, as well as the incidence of neuropathic pain using the validated PainDetect questionnaire [34].

## 4. Socio-Economic Implications of Excision Margins

The implications of surgical decision making extend beyond prognosis, affecting the quality of life of individual patients as well as presenting a socio-economic challenge to healthcare systems globally. Increasing the radial surgical margin from 1 cm to 2 cm increases the size of the resulting defect from 2 cm to 4 cm in diameter. Although this difference may at first seem trivial, it can present a substantial difficulty in repairing the wound, especially if affecting anatomic sites such as the head and neck, where critical structures need to be preserved for both functional and cosmetic considerations, or the distal extremities, where tissue laxity is limiting. It often necessitates the conversion from a simple primary closure to a complex repair involving skin grafts or flaps, which has been reflected in trial data. The Intergroup trial reported 46% of patients treated with 4 cm margins had a skin graft compared with only 11% with 2 cm margins [31]. In the Scandinavian study, for those treated with 2 cm margins, primary closure of the wound was possible in 69% of cases and the use of skin grafts was more frequent in the 4 cm group (12% vs. 47%) [33]. This was similar to the findings of the MelMarT feasibility report, in which over a third of patients in the 2 cm arm required reconstruction with either a skin graft or a local flap, twice that which was required in the 1 cm arm (39.4% vs. 13.6%, respectively; *p* = 0.0001). This difference was even more pronounced in the patients with head and neck melanoma (1 cm: 8.3% vs. 2 cm: 68.8%; *p* = 0.002), although this should be interpreted with caution given the small patient numbers resulting in wide confidence intervals [34]. Given that those with head and neck melanoma have been largely excluded from trials, thus representing < 1% (*n* = 44) of pooled participants to date, it is possible that the true reconstructive burden remains underestimated. Despite substantial heterogeneity between trials, meta-analysis has also found a significant difference in the risk of requiring a skin graft or local flap (RR 0.30; 95% CI 0.19–0.49, *p* < 0.00001) [38]. In two meta-analyses, the estimated number needed to harm (NNH) was three, indicating that for every three patients undergoing a wider excision, one patient would undergo a reconstruction who would otherwise not require it if a narrower margin had been used [36,38].

The increased reconstructive burden associated with wider margins brings with it significant clinical and socio-economic ramifications, including additional risks of morbidity; prolonged hospital stay; disfiguration; chronic pain; and functional loss requiring rehabilitation [25,26,27]. The Intergroup study found that the principal factor that influenced length of hospital stay was the need for skin grafting to close the wound; the hospital stay for those who had a skin graft was 3.5 days longer than for those whose wound was closed primarily (6.5 days vs. 3.0, *p* < 0.01). Balch et al. found that the use of a skin graft was associated with delayed ambulation postoperatively as well as a slightly higher rate of wound infection compared to those who had primary closure (5.7% vs. 2.8%, *p* = 0.07) [26,31]. In the UK MSG trial, there were a greater number of surgical complications in the wider 3 cm arm compared to the 1 cm arm (15% vs. 8%) and the most common complications were partial or complete graft loss (2% in the 1 cm group, 4% in the 3 cm group) and wound dehiscence (2% in both groups) [32,46]. Although the MelMarT-I trial found no significant difference in the overall surgical adverse event rate between the two groups, they did identify a statistically significant increase rate of wound necrosis (including partial/total loss of skin graft) in the wide (2 cm) arm (3.6 vs. 0.5%, *p* = 0.036), which they attribute to the increased rate of reconstruction in the 2 cm arm [34]. When surgical adverse events were explored in a recent meta-analysis, the number of pooled participants was particularly small (862–1762), and they found no significant difference in events between narrower (1–2 cm) and wider margins, specifically for wound dehiscence (RR 0.96; 95% CI 0.54–1.71; *p* = 0.88) and wound infection (RR 1.22; 95% CI 0.68–2.17; *p* = 0.50) [38].

There remains a paucity of quantitative evidence regarding the cost-effectiveness of WLE margins; indeed, none of the published trials or meta-analyses, including the Cochrane review, included any assessment of cost-effectiveness of wider excision margins. An indication of the economic impact of implementing narrower margins can be appreciated from retrospective analysis. In a single-centre UK-based study, 1184 patients diagnosed with pT1b to pT4b primary cutaneous melanoma underwent WLE with either a narrow 1 cm (*n* = 229, 19.3%) or wider 2–3 cm margin (*n* = 995, 80.7%) [27]. In line with trial data, the authors found the odds of needing a reconstruction significantly increased to greater than 3:1 in the wider margin group compared to the narrower margin group. Furthermore, the need for reconstruction significantly increased hospitalisation rates (26.6% vs. 63.0%, OR = 4.7; *p* < 0.0001) collectively and individually for the head and neck (26.8% vs. 53.9%), and upper (18.9% vs. 42.3%) and lower extremities (34.8% vs. 77.3%). The use of the narrower 1 cm margin significantly reduced hospitalisation rates in the upper and lower extremities (7.1% vs. 28.5%; *p* = 0.004, 37.9% vs. 58.5%; *p* = 0.005, respectively). Analysis of resource usage and financial tariff data for 889 patients treated from 2012 onwards found that, in cases where reconstruction was required, there was a significant increase in the mean and median overall procedure cost per patient of £180 (*p* < 0.0001) and £346 (*p* = 0.0004) respectively [27].

The MelMarT-II trial will provide much needed high-quality evidence by incorporating a standalone health economic analysis performed for the UK cohort of patients, determining the cost-effectiveness of implementing a 1 cm compared to 2 cm wide local excision margin for primary invasive cutaneous melanomas, which will be crucial to inform national guidelines. This will be particularly pertinent to policy makers within the UK, as within the centrally funded National Health Service (NHS) there is an ever-increasing demand on limited resources; however, the data it provides will be of interest multi-nationally as melanoma represents an increasingly significant challenge to many different modern healthcare systems.

## 5. Interpretation of RCT Data in the Landscape of Contemporary Melanoma Management

A Cochrane review in 2009 pooled 3297 patients from the then five completed RCTs (WHO, Intergroup, European, Swedish, and UK) found no overall survival advantage for wide excision margins (3–5 cm) compared to narrow margins (1–2 cm), (HR 1.04; 95% CI 0.95–1.15; *p* = 0.400). In the absence of a plausible rationale for narrow margins providing an overall survival benefit, there remains the possibility that margins > 2 cm may result in a small but nevertheless clinically important difference in overall survival (up to a 15% relative reduction in overall mortality), which may have yet to be detected due to insufficient primary data. Small sample sizes have meant that all the trials were underpowered to show the equivalence or non-inferiority of narrow margins compared to wide margins. Accordingly, it was also not possible to produce definitive guidance on the optimal minimum margins by the T stage. The lack of data is further compounded by the absence of any consistent definition for “narrow” (1–2 cm) and “wide” (3–5 cm) excision margins between trials [7,38].

Subsequent meta-analyses have similarly found no statistically significant difference in either survival or recurrence outcomes between narrow and wide excision groups, with the exception of that reported by Wheatley et al. in 2016. They included six RCTs and concluded that narrow margins (1–2 cm) may be harmful compared to wide margins (3–5 cm) given a statistically significant difference in DSS (HR 1.17 95% CI 1.03–1.34, *p* = 0.02). This potential survival disadvantage was not reflected in their summary estimate OS (HR 1.09 95% CI 0.98–1.22; *p* = 0.1) and it is also notable that DSS was only reported in four RCTs (Swedish, Intergroup, UK, and Scandinavian), leading to a substantial risk of selective reporting bias [37]. The most recent meta-analysis, which pooled data from 4579 patients across all seven trials (including the recent MelMarT pilot study) once again found no statistically significant difference in outcomes between narrow (1–2 cm) and wide excision (2–5 cm) for overall death, (RR 1.00; 95% CI 0.93–1.07, *p* = 0.97), death due to melanoma (RR 1.11; 95% CI 0.96–1.28, *p* = 0.16), rates of locoregional recurrence (including LR, in-transit metastasis and regional nodal metastasis individually), or rates of distant metastasis. Once again, they concluded that the analysis was still likely underpowered and thus unable to define the optimal excision margins for primary melanoma, with insufficient information contained within the primary studies to perform any subgroup analysis by BT [38].

Data analysis has been further complicated by the creation of hybrid endpoints in some RCTs, which has led to concerns being raised regarding the interpretation of their data [47]. Specifically, in their long-term analysis of the UK MSG trial, it was proposed that the significant finding of poorer DSS in the 1 cm arm compared to the 3 cm arm was directly attributable to the initial finding of an increased rate of locoregional recurrence associated with the use of a narrower margin. Due to lower than estimated incidence rates of local and in-transit recurrence, the authors combined the rate of nodal recurrence (the incidence of which was over five times greater in both the 1 cm and 3 cm study arms) to create a hybrid endpoint of locoregional recurrence that was defined after the trial began. An alternative explanation for this finding, however, is that the survival disadvantage was due to a higher incidence of biologically more aggressive disease in the narrow margin group that was undetected at the time of the intervention, rather than resulting from the narrow margin intervention itself [46,47]. The possibility that there were between-group differences in rates of SN positivity leading to differences in outcome has been raised and dismissed as unlikely by some, given the degree of protection inherent in the randomisation process as well as the fact that in all of the studies, all other known prognostic characteristics were well-matched between groups [37,46]. However, it is notable that despite randomisation, careful stratification, and well-matched baseline clinicopathological characteristics, in the recent MelMarT feasibility study, the rate of SN positivity was noted to be higher in the 1 cm compared with the 2 cm group (22.9% vs. 15.2%, *p*= 0.058), demonstrating that such a chance imbalance is indeed feasible [34].

This highlights another, and arguably the most clinically significant, limitation in the existing evidence for surgical margins: the absence of SNB as a staging criterion in all but one of the RCTs. This procedure has become standard care in patients with pT2 and above tumours, as a result of the increased accuracy of initial staging it affords, and the fact that, where identified, the presence of regional nodal metastases has been shown to be the single most important independent predictor of both recurrence and survival in patients with intermediate thickness melanoma (1–4 mm) [48,49,50].

There is still a potential therapeutic aspect of SNB that needs to be accounted for when extrapolating evidence to guide modern practice. The second Multicenter Selective Lymphadenectomy Trial (MSLT-II) found that for 823 patients with a positive SN diagnosed and removed at the time of WLE, no nodal recurrence was observed in 80% of patients at 10 years follow-up [51]. If SNB had been employed in previous RCTs, not only would those patients with poorer prognostic disease at the time of wider excision have likely been identified, but the removal of the sentinel node in those patients may have achieved regional nodal disease control in most cases, bringing the rate of the hybrid locoregional recurrence reported in the UK MSG trial into questionable, if any, significance.

Finally, with SNB now representing a gateway to adjuvant systemic therapies with proven survival benefits for those identified with micrometastatic disease, small survival benefits associated with wider margins, if they do exist, may clinically become less important, whilst in parallel, the potential detriment to QOL outcomes becomes unjustifiable.

## 6. Future Directions

### 6.1. WLE in the Management of Rarer Subtypes

The overwhelming majority of RCT evidence gathered to date has been undertaken in Caucasians, with a subsequently high proportion of superficial spreading melanomas. Although the incidence of melanoma is significantly higher in Caucasian populations, its incidence is rising in all. In contrast, less common subtypes, notably acral lentiginous melanoma (ALM) including subungal melanoma (SUM), are the most common subtype of melanoma diagnosed in Fitzpatrick skin types III–VI including people from African, Hispanic, and Asian populations. There are no published RCTs or systematic reviews to define the excision margins specifically for ALM and SUM, and, being excluded from several of the previous RCTs, the generalisability of the recommended margins to patients in these populations is questionable [24,29,34]. The evidence base for the surgical management of these subtypes is therefore derived from predominantly retrospective analysis, which extends beyond the scope of this review. The accrual of further randomised evidence to inform guidelines is required to improve outcomes for these patient groups and tackle the persistent disparity in outcomes between Caucasian and non-Caucasian populations [52].

### 6.2. Will WLE Become Obsolete in the Adjuvant and Neoadjuvant Setting?

It has been argued by some that with the advent of adjuvant and neoadjuvant systemic therapies, WLE may already be obsolete in the contemporary era of melanoma management [53]. Multiple studies have demonstrated that adjuvant therapy improves recurrence-free survival in both stage III and high-risk stage II patients, which may justify the use of narrower excision margins [54,55,56]. It is important to appreciate that the current advances in prognosis have been achieved in the context of a combination of the current surgical management (including the current recommended margins of WLE) and adjuvant therapy. The limited available data on locoregional recurrence show only a modest reduction associated with a high cost and a substantial risk of toxicity; that is, 17% of the pembrolizumab patients in Keynote 716 and 10% of nivolumab patients in CheckMate76K reported grade 3–4 treatment-related adverse events [54,56]. Furthermore, not all patients who are eligible or planned for adjuvant therapy will undergo it due to co-morbidities or concerns regarding toxicities; therefore, optimal surgical management is still of paramount importance [57]. Therefore, at present, there remains a lack of evidence to justify the en bloc reduction in excision margins for those eligible for adjuvant therapy, let alone omission of WLE entirely. The data provided from participants who also undergo adjuvant therapies in MelMarT-II may contribute to further inform the suitability of possible de-escalation of surgery in this patient group in the future.

In contrast, we may see a widespread de-escalation of surgery sooner for those in the neoadjuvant setting, where this approach has been shown to have significant activity in those patients with macroscopic stage III nodal melanoma. The results of the PRADO trial provided an indication that in those patients who achieve a major pathological response, de-escalation of surgery may be safe, but also that escalation of non-responding patients could improve relapse-free survival [58]. Furthermore, available data suggest that disease is unlikely to recur in patients with a good clinical response to neoadjuvant therapy [59]. This may enable the personalisation of excision margins according to the degree of pathological response; some centres have already reported achieving good locoregional disease control in selected cases of patients with melanomas > 2 mm in thickness who underwent wide excision with narrower 1 cm margins following a good clinical response to neoadjuvant therapy [60]. Once again, the introduction of additional risks of immunotherapy-related adverse events must be considered, as they may lead to delays in surgical management. This raises a concern that neoadjuvant therapy may result in a ‘missed opportunity’ in patients who might otherwise have achieved locoregional disease control with surgical management alone; however, the current available evidence does not support this [59].

## 7. Conclusions

For patients diagnosed with cutaneous primary melanoma, and the clinicians who treat them, choosing the optimum WLE margins remains a fundamental decision with significant outcomes at both individual and society levels. It is expected that the results of MelMarT-II will define the surgical treatment internationally for high-risk, AJCC stage II primary cutaneous melanoma patients; the determination to resolve this issue and achieve a definitive answer to this fundamental question after over a century of debate is evident in the fervent engagement of clinicians, patients, and other stakeholders worldwide with the ongoing MelMarT-II trial. Its feasibility findings and progress to date, with over 1600 patients already enrolled, are an encouraging sign that we are on track to achieving this aim.

## Figures and Tables

**Figure 1 cancers-16-00895-f001:**
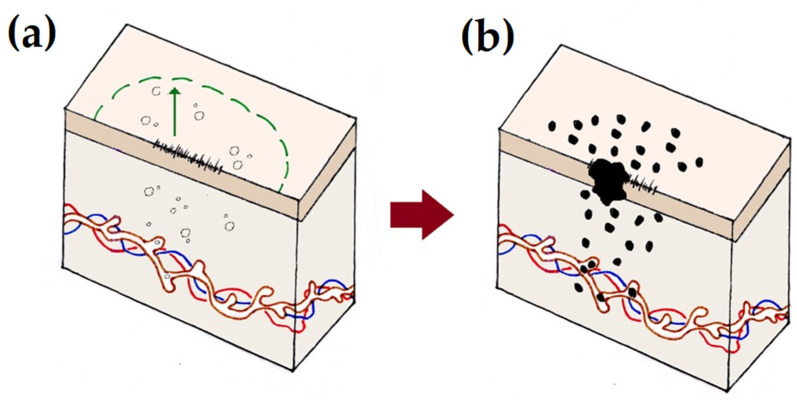
Wide local excision of a primary cutaneous melanoma. Following diagnostic excision, a larger area of tissue surrounding the scar (black dashed line) is excised (**a**), typically down to the level of the fascia using a wide local excision (green dashed line). The width of excision margin chosen (green arrow) is proportional to the Breslow thickness of the primary. Wide local excision aims to remove any local micrometastases or genetically abnormal cells (black outlines) that may still be harbored in the otherwise healthy tissue and/or the superficial lymphatics (brown) and that, if not removed with an adequate margin, may otherwise lead to a locoregional relapse (**b**), manifesting as an in-scar recurrence or in-transit metastases (black).

**Table 1 cancers-16-00895-t001:** A summary of the global guidelines for the recommended clinical excision margins for invasive primary cutaneous melanoma [18,19,20,21,22,23,24].

Breslow Thickness (mm)	UK (2022)	USA (2014)	Australia/New Zealand (2018)	Canada (2017)	Netherlands (2013)	Germany (2014)	Brazil (2015)	Japan (2019)
≤1.0	1 cm	1 cm	1 cm	1 cm	1 cm	1 cm	1 cm	1 cm
1.01–2.00	1–2 cm	1–2 cm	1–2 cm	1–2 cm	1 cm	1 cm	1–2 cm	1–2 cm
2.01–4.00	≥2 cm	2 cm	1–2 cm	2 cm	2 cm	2 cm	2 cm	2 cm
>4.0	≥2 cm	2 cm	2 cm	2 cm	2 cm	2 cm	2 cm	2 cm

**Table 2 cancers-16-00895-t002:** Summary of clinical trials evaluating the width of wider local excision margins for cutaneous primary melanoma.

Trial and Associated Publications [Ref.]	Years of Enrollment	Patient/Tumour Characteristics	Surgical Margins	Sample Size	Outcomes Reported	Length of Follow-Up
**WHO Trial** Veronesi 1988 [28] Veronesi 1991 [39] Cascinelli 1998 [40]	1980–1985	Patients ≤ 65 years with primary melanoma < 2 mm. Excluded melanomas on face, fingers, toes.	1 cm vs. 3 cm	612	OS DFS LR ITM RNM DM	Mean 7.5 years
**European Trial** Khayat 2003 [30]	1981–1986	Patients < 70 years with primary melanoma < 2.1 mm. Excluded toe, nail, or finger lesions.	2 cm vs. 5 cm	337	OS DFS LR RNM DM	Median 16 years
**Swedish Trial I** Ringborg 1996 [29] Cohn-Cedermark 2000 [41]	1982–1990	Primary melanoma > 0.8 mm and ≤2 mm on the trunk or extremities. Excluded melanomas on the head and neck, hands, feet.	2 cm vs. 5 cm	989	OS RFS LR RNM DM	Median 11 years
**Intergroup Trial** Balch 1993 [31] Karakousis 1996 [42] Balch 2001 [26]	1983–1989	Primary melanoma 1–4 mm on the trunk or extremities. Excluded lesions occurring on the head and neck or below the knee or elbow.	2 cm vs. 4 cm	468 (Randomised portion)	OS DFS LR ITM RNM DM Need for skin graft Duration of hospital stay Wound infection Wound dehiscence	Median 10 years
**Swedish Trial II** Gillgren 2011 [33] Utjes 2019 [43]	1992–2004	Patients ≤ 75 years with a primary melanoma ≥ 2 mm on the trunk or extremities. Excluded melanoma of the hands, feet, head and neck, and anogenital region.	2 cm vs. 4 cm	936	OS MSS LR ITM RNM DM	Median 19.6 years
**UK MSG Trial** Thomas 2004 [34] Hayes 2016 [29]	1993–2001	Primary melanoma > 2 mm on the trunk or limbs. Excluded melanoma on the head and neck, soles of feet, or palms of hands.	1 cm vs. 3 cm	900	OS MSS LR ITM RNM DM	Median 8.8 years
**MelMarT Trial** Moncrieff 2018 [34]	2015–2016	Primary melanoma > 1 mm thickness. Excluded melanoma located distal to the metacarpophalangeal joint; on the tip of the nose, the eyelids, or on the ear. Patients ineligible/unable to undergo staging sentinel node biopsy were excluded.	1 cm vs. 2 cm	377	Reconstruction rates Surgical adverse event (wound dehiscence, haematoma, haemorrhage, infection, necrosis) Quality of Life outcomes	1 year
**MelMarT-II Trial**	2019–Ongoing	Primary melanomas pT2b-pT4 (AJCC Stage II)	1 cm vs. 2 cm	2998 (Target)	DFS OS MSS LR DM Health-related quality of life	Ongoing

OS—overall survival, DFS—disease-free survival, MSS—melanoma-specific survival, LR—local recurrence, RFS—recurrence-free survival, DM—distant metastasis, ITM—in-transit metastasis, RNM—regional nodal metastasis, AJCC—American Joint Committee on Cancer.

## Data Availability

No new data were created or analysed in this study. Data sharing is not applicable to this article.

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
