# Peer review of "A Review of Contemporary Guidelines and Evidence for Wide Local Excision in Primary Cutaneous Melanoma Management"

_cancers, 2024, doi:10.3390/cancers16050895_

Round 1

Reviewer 1 Report

Comments and Suggestions for Authors

I believe that this work is an important analysis of existing RCTs assessing the validity of using wide or narrow margins in the treatment of melanoma patients and at the same time an announcement and indication of the direction of the ongoing MelMarT II trial. The authors made a thorough analysis of previous CRTs in this area and paid attention to absence of SNB as a staging criterion in all but one of the RCTs. This study demonstrates the importance of introducing new international recommendations on optimal scar resection margins with the intention of reducing them, as this has significant implications for improving patients' quality of life and reducing medical care costs.

Author Response

Dear Reviewer,

Thank you very much for taking the time to review this manuscript and for your thoughtful comments. We are pleased you feel it is an important analysis of the existing RCT evidence that provides the foundations for the current recommendations for excision margins, and agree on the the importance of introducing new international recommendations in light of the ongoing MelMarT-II trial.

We have revised the manuscript to include a further section with an expanded focus on anticipating changes in the role of WLE in the setting of the advent of adjuvant and neoadjuvant systemic therapies (section 6.2 in the revised manuscript, highlighted) which we hope you feel will enhance its contribution to the field.

Thank you again for your consideration and we look forward to your further feedback.

Yours Faithfully,

Sophie Orme

Reviewer 2 Report

Comments and Suggestions for Authors

I have read this review with great interest, Authors performed a very meaningful analysis of currently available data on the excision margins, however I would like to ask authors to go with more criticism and anticipate some changes in current approach.

I suggest to include some thoughts from paper: Zijlker LP, Eggermont AMM, van Akkooi ACJ. The end of wide local excision (WLE) margins for melanoma ? Eur J Cancer. 2023 Jan;178:82-87. doi: 10.1016/j.ejca.2022.10.028. Epub 2022 Nov 2. PMID: 36423526.

The retrospective study showed no difference in OS after correction for confounding factors. Pathology studies showed a low incidence of residual melanoma in WLE specimen. Mohs surgery does not show a difference in recurrence rates or OS. WLE is associated with considerable postoperative morbidity, which increases with wider excision margins.

Author Response

Dear Reviewer

Thank you very much for taking the time to review this manuscript and for your thoughtful comments. Please find the detailed responses to these below and the corresponding revisions highlighted in the re-submitted file:

  • “I suggest to include some thoughts from paper: Zijlker LP, Eggermont AMM, van Akkooi ACJ. The end of wide local excision (WLE) margins for melanoma ? Eur J Cancer. 2023 Jan;178:82-87. doi: 10.1016/j.ejca.2022.10.028.”

&

  • “I would like to ask authors to go with more criticism and anticipate some changes in current approach”

Thank you for bringing our attention to the above article which we read with interest. The advent of systemic therapy has certainly led to polarisation of opinions with regard to the role of WLE in contemporary management, which is certainly a factor as to why we think this manuscript is a significant and timely contribution to the field. We agree that this change in landscape is important, but at this early stage much of the debate is rooted in theory in the context of limited published evidence. As such we have cited the article you kindly suggested [53] and expanded the future directions section to include our anticipation of how the role of WLE in this setting may change in the future based on the limited current published evidence (paragraph 6.2 in the revised manuscript, highlighted). We are unable to go with more criticism as you suggest as we disagree with the authors’ conclusions that WLE can be considered obsolete at this time; to briefly summarise the key evidence supporting this conclusion as outlined in the manuscript:

  • Work conducted by North et. al [12] showing the presence of field cells which may represent a plausible mechanism for local persistence. Current pathology reports are unable to detect these.
  • Findings from the MIA working groups that in melanomas ≤ 2 mm thick, histologic margins < 8 mm were associated with a higher rate of local recurrence, whereas histologic margins <16 mm in those with tumors > 4mm thick had worse local disease-free survival (HR, 2.41; P = 0.01), supporting the above [14-16] (addition in the revised article, highlighted lines 95-102)
  • The near significant finding for poorer DSS for the 2 cm excision group vs 4 cm in the Intergroup trial [31], the near significant increase in LR rate in the 2cm group vs the 4 cm group in the Scandinavian trial [33] and the statistically significant difference in the locoregional recurrence rate in the UK MSG trial [32]

Thank you again for your consideration and we look forward to your feedback of the revised manuscript

Yours faithfully,

Sophie Orme

Reviewer 3 Report

Comments and Suggestions for Authors

Since this is a literature review, it would have been appropriate to clearly define the keywords of the search first of all

The most cited works are limited to literature from Northern European countries, neglecting those of other working groups

To be a review of an important and increasingly widespread pathology, it is necessary to extend the analysis to other works with long follow-ups (at least 10 years) and to search for them in the literature

The title should be changed to also include the review or it should be an appendix of the first frequently cited work of MELMAR1

I believe that the number of citations should be implemented, also taking into account the geographical areas to which the authors belong, as it is a very interesting topic

Author Response

Dear Reviewer,

Thank you very much for taking the time to review this manuscript and for your thoughtful comments. Please find the detailed responses to these below and the corresponding revisions highlighted in the re-submitted file:

1) “Since this is a literature review, it would have been appropriate to clearly define the keywords of the search first of all”

Thank you for highlighting this. A “Literature Search & Selection” paragraph (section 2 in the revised manuscript, highlighted) has now been added clearly defining the search strategy.

2) “The most cited works are limited to literature from Northern European countries, neglecting those of other working groups” &

3) “To be a review of an important and increasingly widespread pathology, it is necessary to extend the analysis to other works with long follow-ups (at least 10 years) and to search for them in the literature”

Thank you for your suggestions. With the intention of keeping the article length palatable for the readership, we have taken the decision to comprehensively  evaluate the highest levels of evidence currently available, limiting the search to RCTs and systematic reviews. We recognise there is an inherent bias in the current literature resulting from the inclusion of a large majority of Caucasian participants in previous trials, and the citations similarly reflect this, predominantly citing working groups from Europe, Northern America, Canada and Australasia, with many of our citations representing multinational collaborations between these groups. As such we respectfully disagree with your comment that the most cited works are limited to literature from Northern European countries.

To our knowledge there are no published RCTs or systematic reviews to define the excision margins specifically for ALM and SUM which are more common in non-Caucasian populations. With your suggestions in mind, we have briefly but explicitly stated this bias in the revised manuscript (section 6.1, highlighted) as we agree this an important future avenue for further research.

We agree that a contemporary review of this nature should endeavour to represent the global scale of the problem, and this is reflected by the summary of current recommendations in Table 2 which are taken not only from the guidelines of Northern European countries but also from those of the USA, Australia/New Zealand, Canada, Brazil and Japan (the latter added in the revised article, highlighted in yellow). 

4) “The title should be changed to also include the review or it should be an appendix of the first frequently cited work of MELMAR1”

Thank you for your suggestion, we have amended the title to more clearly identify the nature of the article as a narrative review

5) “I believe that the number of citations should be implemented, also taking into account the geographical areas to which the authors belong, as it is a very interesting topic”

Please refer to our response to your comments numbered 2&3.

Thank you again for your consideration and we look forward to your feedback on the revised manuscript

Your faithfully,

Sophie Orme

Round 2

Reviewer 3 Report

Comments and Suggestions for Authors

All recommendations have been implemented, well done , I really appreciated the precision